# The 419th Aspartic Acid of Neural Membrane Protein Enolase 2 Is a Key Residue Involved in the Axonal Growth of Motor Neurons Mediated by Interaction between Enolase 2 Receptor and Extracellular Pgk1 Ligand

**DOI:** 10.3390/ijms251910753

**Published:** 2024-10-06

**Authors:** Bing-Chang Lee, Jui-Che Tsai, Yi-Hsin Huang, Chun-Cheng Wang, Hung-Chieh Lee, Huai-Jen Tsai

**Affiliations:** 1Department of Life Science, Fu Jen Catholic University, New Taipei City 242062, Taiwan; bingchanglee@ntu.edu.tw (B.-C.L.); sophiahuang990326@gmail.com (Y.-H.H.); d91243003@ntu.edu.tw (H.-C.L.); 2Graduate Institute of Applied Science and Engineering, Fu Jen Catholic University, New Taipei City 242062, Taiwan; juichetsai1992@ntu.edu.tw; 3Institute of Molecular and Cellular Biology, National Taiwan University, Taipei 10617, Taiwan; r12b43006@ntu.edu.tw

**Keywords:** receptor, extracellular, neurite outgrowth, p-Cofilin, siRNA

## Abstract

Neuron-specific Enolase 2 (Eno2) is an isozyme primarily distributed in the central and peripheral nervous systems and neuroendocrine cells. It promotes neuronal survival, differentiation, and axonal regeneration. Recent studies have shown that Eno2 localized on the cell membrane of motor neurons acts as a receptor for extracellular phosphoglycerate kinase 1 (ePgk1), which is secreted by muscle cells and promotes the neurite outgrowth of motor neurons (NOMN). However, interaction between Eno1, another isozyme of Enolase, and ePgk1 failed to return the same result. To account for the difference, we constructed seven point-mutations of Eno2, corresponding to those of Eno1, and verified their effects on NOMN. Among the seven Eno2 mutants, *eno2*-siRNA-knockdown NSC34 cells transfected with plasmid encoding the 419th aspartic acid mutated into serine (Eno2-[D419S]) or Eno2-[E420K] showed a significant reduction in neurite length. Moreover, the Eno2-ePgk1-interacted synergic effect on NOMN driven by Eno2-[D419S] was more profoundly reduced than that driven by Eno2-[E420K], suggesting that D419 was the more essential residue involved in NOMN mediated by Eno2-ePgk1 interaction. Eno2-ePgk1-mediated NOMN appeared to increase the level of p-Cofilin, a growth cone collapse marker, in NSC34 cells transfected with Eno2-[D419S] and incubated with ePgk1, thereby inhibiting NOMN. Furthermore, we conducted in vivo experiments using zebrafish transgenic line *Tg(mnx1:GFP)*, in which GFP is tagged in motor neurons. In the presence of ePgk1, the retarded growth of axons in embryos injected with *eno2*-specific antisense morpholino oligonucleotides (MO) could be rescued by *wobble-eno2*-mRNA. However, despite the addition of ePgk1, the decreased defective axons and the increased branched neurons were not significantly improved in the *eno2-[D419S]-*mRNA-injected embryos. Collectively, these results lead us to suggest that the 419th aspartic acid of mouse Eno2 is likely a crucial site affecting motor neuron development mediated by Eno2-ePgk1 interaction, and, hence, mutations result in a significant reduction in the degree of NOMN in vitro and axonal growth in vivo.

## 1. Introduction

Enolase, a critical glycolytic enzyme, catalyzes the conversion of 2-phosphoglycerate into phosphoenolpyruvate, which plays a pivotal role in ATP production during glycolysis. Nevertheless, Enolase is implicated in ischemia, hypoxia, and various metabolic and neurodegenerative diseases [1,2,3,4,5,6,7]. Although predominantly localized in the cytoplasm, Enolase also appears in the nucleus and cell membrane, suggesting that enolases possess diverse cellular functions. For example, upon specific signaling stimuli, Enolase translocates to the cell surface where it is implicated in inflammatory responses and extracellular matrix degradation [8]. 

The Enolase family consists of three isozymes: the ubiquitously expressed Eno1 (α-enolase), neuronal Eno2 (γ-enolase), and muscle-specific Eno3 (β-enolase). They share 79% identity and 84% similarity in their amino acid sequences [9,10]. In general, Enolase forms dimers through hydrophobic noncovalent interactions and distributes at different tissues: αα dimer in most mature tissues, αγ and γγ dimers in neural tissue, and αβ and ββ dimers in muscular tissue [9]. Eno1 is found on the cell membranes of neurons, monocytes, and endothelial cells, serving as a plasminogen receptor [11,12]. Eno1 binds with plasminogen at the cell surface, subsequently generating plasmin by cleavage of specific proteases. Plasmin plays roles in extracellular matrix degradation and tumor migration [13]. Eno2 can be detected in the blood and cerebrospinal fluid, serving as a potential biomarker for assessing neuronal damage and prognosis following CNS injuries [7,14,15,16,17,18,19]. Eno2 can also act as a neurotrophic factor, protecting neuronal survival, differentiation, and neurogenesis in Alzheimer’s disease [6,20]. More importantly, the C-terminus of Eno2 plays a crucial role in neurological diseases. Moreover, Eno2, also known as neuron-specific enolase (NSE), is primarily distributed in the central and peripheral nervous systems and neuroendocrine cells. Eno2 is observed on the cell membranes of rat brain synaptic terminals and NSC34 neural cells [21,22]. The hydrophobic region at the N-terminus of Eno2 attaches to the cell membrane, while the C-terminal domain is involved in signaling, mediating actin cytoskeletal reorganization, neural outgrowth, and promoting neuronal survival, differentiation, and axonal regeneration through the PI3K/AKT, MAPK/ERK, and RhoA kinase pathways [23]. It has been further reported that the translocation of Eno2 to the cell membrane can be fostered through interaction between γ-syntrophin and its PDZ-binding motif formed by Ser-Val-Leu at residues 432–434 of the C-terminus [24,25]. Interestingly, administration of the C-terminal 30 amino acid-long peptide of γ-Enolase could promote survival of cerebral cortical embryonic cells of rat, suggesting the functional importance of Eno2 C-termini on the cell membrane [26]. Lastly, Eno3 is primarily located in smooth muscle cells where it binds with actin thin filaments to produce ATP for muscle contraction [27]. Besides skeletal and cardiac muscles, Eno3 is also found in the liver and lungs [28]. Lin et al. [29] discovered that extracellular administration of phosphoglycerate Kinase 1 (Pgk1) to NSC34 neural cell culture significantly promotes axonal growth via a newly identified signaling pathway. Specifically, extracellular Pgk1 (ePgk1) reduces phospho-Cofilin at S3 (a growth cone collapse marker) through inhibition of Rac1-GTP/p-Pak1-T423/p-p38-T180/p-MK2-T334/p-Limk1-S323 signaling, resulting in the promotion of neurite outgrowth of motor neurons (NOMN). This regulatory pathway is distinct from the well-known intracellular Pgk1 role in glycolysis, which supplies ATP for axonal growth. Indeed, intramuscular injection of Pgk1 was shown to significantly reduce or delay neuromuscular junction (NMJ) denervation in Rtn4al-overexpressing Amyotrophic lateral sclerosis (ALS)-like zebrafish and SOD1-mutated ALS-like mice, leading to improved muscle contraction and mobility [29,30]. Recently, Fu et al. [22] reported that Eno2 on the cell membrane of motor neurons acts as a receptor for ePgk1 to promote NOMN. This finding was based on a Co-IP study demonstrating that Eno2 effectively binds with Pgk1-Flag and vice versa, suggesting that ePgk1 could ligate with membranous Eno2 receptors, as further confirmed by direct observation from immune-electron microscopy. Further, they propose that the 325th to 417th amino acids of ePgk1 interact with the 404th to 431st amino acids, reducing p-Cofilin-S3 and promoting axonal growth. They also demonstrated that co-overexpression of ePgk1 and Eno2 resulted in synergism that increased NOMN compared to overexpression of either ePgk1 or Eno2 alone. Interestingly, although Eno1 and Eno2 have very similar structures, ePgk1 binds only with Eno2, not Eno1 [22]. In the present study, we set out to solve this intriguing conundrum, first asking whether elucidating the difference in NOMN growth between ePgk1 and mutated Eno2 would lead to the discovery of critical amino acid(s) of Eno2 associated with improved NOMN. Our search began with a screening and comparison of 30 amino acids in the C-termini of Eno1 and Eno2, excluding the three amino acids of the PDZ domain, leaving a difference of seven unique amino acids. To quickly verify the key amino acid controlling the development of motor neurons through ePgk1-Eno2 interaction, we performed both in vitro and in vivo studies, and collectively demonstrated that the 419th aspartic acid of Eno2 plays an important role in Eno2-ePgk1-mediated NOMN. A plausible docking model showing why the 419th aspartic acid of Eno2 is indispensable to Eno2-ePgk1-mediated NOMN was also proposed.

## 2. Results

### 2.1. Effects of Overexpression of Mutated Eno2 on Promoting the Axonal Growth of Motor Neurons 

It is well known that Eno2 can interact with ePgk1 to promote NOMN, whereas interaction between Eno1, another isozyme of Eno, and ePgk1 failed to return the same result. To account for the difference, we employed a PCR-directed mutagenesis strategy (Figure 1) using the flanking and mutagenic primers (Table 1) to generate plasmids harboring seven point-mutations of Eno2, corresponding to those of Eno1, including Eno2-[L401I], -[M411L], -[D419S], -[E420K], -[R422K], -[H426R], and -[N427S]. Initially, to determine if overexpression of these Eno2 mutants would affect NOMN, we cultured NSC34 cells on serum deprived medium (SDM) with 2% fetal bovine serum (FBS) for five days. We then used the neurite length derived from those differentiated cells as a NOMN determinant. We used *eno2*-specific siRNA to knock down endogenous Eno2. For the rescue experiment, we engineered plasmid pCS2 harboring a wobble-modified *eno2* cDNA, which contains the modified nucleotide at the wobble (wb) position of the codon without changing the encoded amino acids of Eno2 (Eno2-wb). This modification prevents complete complementarity of base pairs with introduced siRNA. NSC34 cells were first transfected with siRNA-containing plasmid, followed by individual transfection with plasmid containing *eno2-wb* (as the control group) or each of the mutated *eno2*, as noted above, and we then quantified the resultant average length of NOMN. First, Western blot analysis, as shown in Figure 2, demonstrated that the protein level of endogenous Eno2 was dramatically reduced in the siRNA-treated cells (lane 3), while the Eno2 level could be rescued up to the control level, set as 1 (lane 1), in the siRNA-treated cells transfected with plasmid DNA encoding *eno2-wb* (lane 4).

Next, we examined and quantified the neurite length developed from the differentiated NSC34 cells (Figure 3A,B). As shown in Figure 3C, compared to the control group rescued by Eno2-wb, set as 1, the results demonstrated the following effects on NOMN: (1) neurite growth was retarded in cells overexpressed with Eno2-wb-[D419S], -[E420K], and -[M22K] mutants; (2) neurite growth was significantly improved in cells overexpressed with Eno2-wb-[L401I] and -[N427S] mutants; and (3) neurite length remained unchanged in cells transfected with Eno2-wb-[M411L] and -[H426R] mutants. Since the effects of mutant Eno2 overexpression on NOMN were determined by neurite length, it could be inferred that retarded development of NOMN by mutant Eno2 overexpression might suggest Eno2 loss of function, in turn suggesting that a specific original amino acid is a key player. Therefore, to determine the key amino acid of Eno2 involved in the Eno2-ePgk1 interaction, we chose Eno2-[D419S] and -[E420K] mutants, both of which caused retarded NOMN. The Eno2-[M411L] mutant, which caused no change in NOMN, served as a parallel reference to further analyze the effects on NOMN in the presence of ePgk1. 

### 2.2. The Synergic Promotion of Axonal Growth Derived from Cells Transfected with Mutated Eno2 Plus Pgk1 Immersion

To identify the key amino acids involved in Eno2-ePgk1 interaction in the promotion of NOMN, we added exogenous recombinant Pgk1-Flag fusion protein to the culture medium. As shown in Figure 4, our findings revealed that: (1) the neurite length of siRNA-transfected cells (indicated as siRNA+, Pgk1-, and pCS2 vector) was shorter than that of cells rescued by Eno2-wb, which was a modified Eno2 without siRNA interference for translation (indicated as siRNA+, Pgk1-, and Eno2-wb) (lane 2 vs. 4), suggesting that retarded neurite growth caused by Eno2-knockdown could be rescued by introducing Eno2-wb; (2) the neurite length of siRNA-transfected cells (indicated as siRNA+, Pgk1+, and pCS2 vector) was significantly shorter than that of cells rescued by Eno2-wb (indicated as siRNA+, Pgk1+, and Eno2-wb) (lane 1 vs. 3), suggesting that the addition of Pgk1 could improve neurite length of siRNA-treated cells rescued by Eno2-wb in a synergic manner compared to siRNA-treated cells in which Eno2 was knocked down. Compared to siRNA-treated cells rescued by Eno2-wb, set as 1 (lane 1), the neurite length of *eno2-wb-[M411L]*-transfected cells remained unchanged in the presence of Pgk1 (lane 5). However, the neurite length of siRNA-treated cells transfected with *eno2-wb-[D419S]* and *–[E420K]* in the presence of Pgk1 exhibited 0.73- and 0.85-fold decreases in NOMN growth, respectively (lanes 6 and 7), indicating that the synergic improvement of NOMN resulting from Eno2-Pgk1 interaction was not favorable to introducing either mutant *eno2-wb-[D419S]* or *-[E420K]*. This line of evidence suggests that amino acids at positions 419 and 420 of Eno2 might play a critical role in NOMN mediated by Eno2-ePgk1 interaction.

### 2.3. Axonal Defect That Occurred in Eno2-Knockdown Zebrafish Embryos Was Rescued by Introducing eno2 mRNA Mutant

The above in vitro experiment demonstrated that the reduction in neurite outgrowth could be attributed to Eno2-[D419S] and -[E420K] mutants; therefore, we explored whether a similar effect would occur in vivo. To make this determination, we studied axonal development in the trunk of motorneuron-GFP-tagged transgenic zebrafish (Figure 5A). First, we knocked down endogenous Eno2 using *eno2*-specific antisense morpholino oligonucleotides (MO) *eno2*-specific MO (Figure 5B), followed by rescue with microinjection of mRNAs encoding these two Eno2 mutants. Based on the axonal phenotypes shown in the trunk of 30 hpf embryos, we categorized two groups: one was normally developed axons (Figure 5C), the other was defectively developed axons, including mild and severe defects (Figure 5D,E). The occurrence rate of defective development of axons was then calculated. Compared to the non-injected group (20%), the occurrence rate of shortened neurites was increased in the MO-injected group (59.3%), while that in the MO plus *eno2-wb*-mRNA-injected group was decreased (33.3%). However, compared to the MO plus *eno2-wb*-mRNA-injected group, the occurrence rate of shortened neurites was significantly increased in the MO plus *eno2-wb-[D419S]*-mRNA-injected group (51.1%) (Figure 5F), suggesting that the rescue potential against shortened neurite defects was significantly reduced if embryos were introduced by mRNA coding for Eno2 S419 mutated from D419. 

Using a similar strategy, we studied the neurite defect of Eno2-knockdown zebrafish embryos rescued by introducing *eno2-wb-[E420K]* mRNA. Similarly, compared to the non-injected group (12.8%), the occurrence rate of shortened neurites was increased in the MO-injected group (40.4%), while it was reduced in the MO plus *eno2-wb*-mRNA-injected group (24.3%). We found that the occurrence rate of shortened axons in the MO plus *eno2-E420K*-mRNA-injected group (40%) was not significantly different when compared with the MO plus *eno2-wb*-mRNA-injected group (Figure 5G), suggesting that Eno2-K420 exerted no significant impact on the rescue of shortened neurite defects. 

### 2.4. Effect of Overexpressing Mutated Eno2 on the Improvement of Branched Neuron Formation in Zebrafish Embryos

Next, to investigate whether the effect of mutations at positions 419 and 420 of Eno2 on Eno2-ePgk1 interaction, we overexpressed mouse *eno2-wb-[D419S]* mRNA in zebrafish embryos, followed by immersion of recombinant Pgk1 to quantify the number of forming branched neurons shown in the trunk of embryos (Figure 6A,B). The results showed that the occurrence rate of branched neurons formed in the embryos injected with *eno2-wb* mRNA increased compared to the non-injected control group, with rates of 58.7% and 41.3%, respectively (Figure 6C). However, the occurrence rate of branched neurons formed in the *eno2-wb-[D419S]-*mRNA-injected embryos was significantly decreased compared to that of *eno2-wb*-injected embryos, with rates of 39% and 58.7%, respectively (Figure 6C), suggesting that mutated D419S could significantly reduce the improvement of branching NOMN mediated by Eno2-ePgk1 interaction.

Using a similar strategy, we studied whether branched neuron formation of zebrafish embryos could also be affected by injection of *eno2-wb-[E420K]* mRNA. We found results similar to those from the *eno2-wb-[D419S]*-mRNA-injected embryos, with the occurrence rate of branched neuron formation of the *eno2-wb*-mRNA-injected group increasing compared to the non-injected control group, with rates of 54% and 38.2%, respectively (Figure 6D). Interestingly, unlike the *eno2-wb-[D419S]-*mRNA-injected embryos, the occurrence rate of branched neuron formation between the *eno2-wb-[E420K]*-mRNA-injected embryos and the *eno2-wb*-mRNA-injected embryos was not statistically significant (Figure 6D). This line of evidence suggests that the impact of the mutated E420K on the improvement of neuron branching was neither greater nor lesser than that of the mutated D419S. Nonetheless, when considering both in vitro and in vivo results, we concluded that D419 of Eno2 is an important amino acid residue that plays a more critical role than E420 in the improvement of branching NOMN mediated by Eno2-ePgk1 interaction.

### 2.5. Expression Level of Phosphorylated Cofilin (p-Cofilin) within NSC34 Cells

Since the above experiments supported our hypothesis that D419 of Eno2 might play a more critical role in the improvement of NOMN growth and branching mediated by Eno2-ePgk1 interaction, we continued to investigate the level of p-Cofilin, a growth cone collapse marker, within the Eno2-wb-[D419S] cells. Total proteins were extracted from siRNA-treated NSC34 cells transfected with D419S-containing DNA with or without addition of Pgk1, followed by performing Western blot analysis to detect the expression level of p-Cofilin. As shown in Figure 7, compared to the cells transfected by vector pCS2 harboring DNA fragment transcribed into siRNA, which served as control and was set as 1 (lane 1), the intensity of the p-Cofilin level extracted from siRNA-treated cells rescued by Eno2-wb was reduced to 0.84 (lane 2), suggesting that the improvement of neurite outgrowth was caused by the rescue of Eno2-wb. Moreover, the intensity of the p-Cofilin level extracted from siRNA-treated cells supplied with recombinant Pgk1 was further decreased to 0.77 (lane 3), suggesting that extracellular addition of Pgk1 in the Eno2-knockdown cells could also further improve neurite growth, since endogenous Eno2 was not completely absent in the siRNA-treated cells, as demonstrated in Figure 2. Interestingly, the intensity of the p-Cofilin level extracted from siRNA-treated cells rescued by DNA encoding Eno2-wb and supplied with recombinant Pgk1 exhibited a further decrease, down to 0.68 (lane 4), suggesting that neurite growth was enhanced by Eno2-Pgk1 interaction. However, unlike the groups of siRNA-treated cells rescued by Eno2-wb and supplied with recombinant Pgk1, the intensity of the p-Cofilin level extracted from siRNA-treated cells rescued by Eno2-wb-[D419S] and supplied with recombinant Pgk1 was increased up to 0.9 (lane 5), suggesting that overexpression of mutant Eno2-wb-[D419S] could lead to decreased neurite growth, even in the presence of Pgk1. This evidence provides additional support to our hypothesis that the point mutation of amino acid at D419 to S419 of Eno2 significantly affects the neurite growth mediated by the Eno2-Pgk1 interaction. 

### 2.6. Docking Model Illustrates That D419 of Eno2 Is a Key Amino Acid Involved in Eno2-Pgk1 Interaction 

This study has established that D419 of Eno2 is a key amino acid and that Eno2-Pgk1 interaction fails to rescue the Eno2-[D419S] mutant. To construct a model to illustrate these findings, we performed a molecular docking analysis of the interaction between Eno2 (PDB ID: 5TD9, orange) and Pgk1 (PDB ID: 2ZGV, blue). Based on the docking model shown in Figure 8A,B, we found that D419 and E420 residues of Eno2 (yellow region) were two amino acids located on the surface of Eno2 close to the interface between Eno2 and Pgk1. Additionally, D419 and E420 of Eno2, two negatively charged amino acids, were the most adjacent residues able to interact with the 325th–417th domain of Pgk1, a positively charged segment forming an α-helix secondary structure (Figure 8A). It also showed that the surface-exposed amino acids of Pgk1 near the Eno2 region included E346, A349, K353, and D357 (Figure 8A), and that the interacting counterpart amino acids of Eno2-D419 and -E420 were Pgk1-K353 and -F346, respectively (Figure 8A). Interestingly, compared to the Eno2-[D419S] interacted with Pgk1-[K353], the side chain of mutated Eno2-S419 was altered (highlighted in yellow, Figure 8A vs. Figure 8B), suggesting the surface charge attraction between Eno2 and Pgk1 might become weaker. 

When we analyzed the polarity of Eno2, we found that both D419 and S419 (indicated with yellow dotted circles) were hydrophilic (Figure 8C,D), suggesting no difference in hydrophilicity between Eno2-D419 and -S419. However, chargeability analysis showed that the aspartic acid at the 419th position of Eno2 presented a negative charge, as indicated by red in Figure 8E, while the serine at the 419th position of Eno2 presented a neutral charge (Figure 8F). Taken together, these docking models could also strengthen our hypothesis that the D419 of Eno2 is a critical residue to improve NOMN mediated by Eno2-Pgk1 interaction.

## 3. Discussion

Both Eno1 and Eno2 polypeptides consist of 434 amino acid residues, sharing 84% similarity [31]. Moreover, both Eno1 and Eno2 polypeptides are composed of 20 α-helices and 12 β-sheets, displaying almost the same tertiary structures [31]. Specifically, the C-terminal 30 amino acid peptide of Eno2 has been proven to be a functional domain that plays a vital role in neural survival [26]. Recently, Fu et al. [22] proposed that the 404th to 431st amino acids of Eno2 interact with the 325th to 417th amino acids of ePgk1 to reduce phosphorylated Cofilin at Ser3 and thereby promote axonal growth in a synergic manner. After we screened and compared 30 amino acid residues in the C-termini of Eno1 and Eno2, excluding a PDZ-binding motif (431–433: S-V-L), which interacts with other PDZ domain-containing proteins involved in intracellular redistribution of molecules and signaling pathways [25], only seven unique amino acids were left. However, ePgk1 only interacts with Eno2, not Eno1, to promote increased NOMN [22]. Considering their sequence homology, this seeming inconsistency prompted us to investigate the possible existence of a unique amino acid residue of Eno2 that is somehow involved in the development of NOMN mediated by ePgk1-Eno2 interaction. Therefore, in the first step of this study, we converted these seven amino acids of Eno2 into those of Eno1 by point-mutation genetic engineering (i.e., site-directed mutagenesis), resulting in Eno2-[L401I], -[M411L], -[D419S], -[E420K], -[R422K], -[H426R], and -[N427S]. Using NSC34 neuronal cells, we examined the effect of overexpressing each mutated Eno2 on NOMN. We found that NOMN was retarded in Eno2-knockdown cells transfected with Eno2-[D419S] and -[E420K] mutants. Moreover, when ePgk1 was added, no synergic improvement in NOMN growth was detected in Eno2-knockdown NSC34 cells transfected with Eno2-[D419S] upon ePgk1-Eno2 interaction when compared to that of cells transfected with Eno2-[E420K]. 

To further examine this discovery in an in vivo system, we used zebrafish embryos and demonstrated that (1) injection of *eno2-[D419S]* mRNA could not significantly rescue the axonal defect that occurred in Eno2-knockdown embryos compared with those embryos injected with *eno2-[E420K]* mRNA (Figure 5) and (2) the number of branched neurons typical of normal axonal growth was more significantly reduced in the Eno2-knockdown embryos injected with *eno2-[D419S]* mRNA, even in the presence of Pgk1, when compared with that of embryos injected with *eno2-[E420K]* mRNA (Figure 6). From this line of evidence, it was concluded that (1) the aspartic acid at position 419 of Eno2, not the glutamic acid at position 420 of Eno2, is involved in NOMN mediated by ePgk1-Eno2 interaction, and (2) among the seven unique amino acids between Eno1 and Eno2, the D419 residue of Eno2 was found to be critical in controlling NOMN mediated by ePgk1-Eno2 interaction. 

### 3.1. A Docking Model Illustrates How the 419th Aspartic Acid of Eno2 Is a Key Residue Involved in NOMN Mediated by Eno2-Epgk1 Interaction

The docking model presented in Figure 8 illustrates that the 325–417 domain of Pgk1 forms an α-helix structure in which the 350–353 region displays a positively charged surface. On the other hand, the 414–420 domain of Eno2 is composed of amino acids with negatively charged side chains, as described by Kang et al. [32]. Additionally, a loop structure is predicted to form between the G418, D419, and E420 residues of Eno2 with the negatively charged D419 residue located at the loop’s apex (Figure 8A,B). Lysine (K), a positively charged residue at position 353 of Pgk1, is supposed to be the nearest counterpart amino acid interacting with Eno2 D419, with a predicted distance between them of 2.63 Å. Such proximity between oppositely charged surfaces on the proteins generates an electrostatic attraction [33], which, in turn, promotes NOMN mediated by Eno2-ePgk1 interaction. However, if Eno2 D419 were mutated into a neutral, polar serine (Eno2 S419), as presented in Eno1 at 419, the electrostatic attraction between ePgk1 and Eno2 would be reduced, leading to increased p-Cofilin (Figure 7), in turn reducing neurite growth and axonal branching demonstrated in vitro and in vivo, respectively (Figure 3, Figure 5 and Figure 6). This reasoning explains why the interaction between Eno1 and ePgk1 failed to improve NOMN.

Similarly, the Eno2 residue at position 420, located below the loop structure, is a negatively charged glutamic acid (E420). The nearest counterpart amino acids involved in Eno2-Pgk1 interaction are predicted to be a positively charged K353 with a distance of 6.00 Å, and a negatively charged E346 with a distance of 6.47 Å (Figure 8A) at Pgk1. Since Eno2-E420 is simultaneously influenced by these two closely positioned residues located at Pgk1, including a negatively charged Pgk1-E346, the electrostatic interaction between the Eno2-E420 and Pgk1 becomes weaker than that of Eno2-D419 and Pgk1 described above. Although theEno2 E420K mutation increases the electrostatic attraction with Pgk1 E346, this electrostatic force is distracted by Pgk1 K353 owing to charge repulsion between the positively charged residues. This explains why the mutant Eno2 E420K exerts less pronounced effect over neurite length, axonal defect, and axonal branches in the rescue experiments when compared to that of mutant Eno2 D419S in this study. The situation with Eno2 R422 is similar to that of D419 since arginine (Arg, R) at position 422 is positively charged and positioned near E346 on Pgk1, thereby providing the necessary electrostatic force for Pgk1-Eno2 binding, consistent with the results shown in Figure 3C.

Analysis of the Eno2-Pgk1 docking model presented here shows that the 405–414 domain of Eno2 does not closely interact with ePgk1, suggesting that Eno2-L410 and -M411 may not be involved in the Eno2-ePgk1 interaction. This observation may explain why the L410I and M411L mutants have no effect on the promotion of NOMN mediated by Eno2-ePgk1 interaction. 

Based on docking model, electrical charge attraction, rather than polarity, between Eno2-D419 and ePgk1-K353 is a critical factor affected NOMN growth mediated by Eno2-ePgk1 interaction. Apart from the attraction of opposite electrical charges between two side chains of counterpart amino acids, other factors near and within the entire microenvironment between two proteins may also be involved in the Eno2-ePgk1 interaction. For example, hydrophobic interactions, salt bridge formation owing to electrostatic forces, van der Waals forces and other factors may also be involved in Eno2-ePgk1 interaction. Therefore, further investigation is required.

### 3.2. Distinct Interaction between Ligands and Receptor Eno Family Depends on the Specific Amino Acid Residues of Receptors Recognized by Ligands 

Miles et al. [34] synthesized a 16-amino acid peptide representing the C-terminal 419th~434th amino acids of Eno1 that could enable plasminogen binding to U937 cells, suggesting that the 16-amino acid peptide at the C-terminus of Eno1 could specifically interact with plasminogen. On the other hand, Hattori et al. [26] reported that the synthetic 30-amino acid peptide corresponding to the 404th–433rd amino acid residues at the C-terminus of Eno2 promotes neuronal survival. This line of evidence suggests that the loop composed of the 419th~422nd amino acids between α19 and α20 helices in both Eno1 and Eno2 is an important motif that determines whether Eno1 and Eno2 could exclusively interact with plasminogen and Pgk1, respectively, resulting in displaying its own distinct function. Interestingly, proteins that could associate with Eno1 are not necessarily associated with Eno2 or vice versa. For example, interaction between extracellular plasminogen and membranous Eno1 depends upon plasminogen recognizing lysine (K) at 420, 422, and 434 at the C-terminus of Eno1 [12,34], which results in plasminogen only interacting with Eno1, not Eno2 [35]. Plasminogen interacting with Eno1 at the K420, K422, and K434 of Eno1 would result in increased p-Cofilin at S3 (a growth cone collapse marker), in turn inhibiting NOMN [36]. More recently, Fu et al. [22] demonstrated that ePgk1 could interact with membranous Eno2, not Eno1, to reduce p-Cofilin through inhibition of p-Pak1/p-p38/p-MK2/p-Limk1 signaling, resulting in the promotion of NOMN. In this study, we are the first to demonstrate that the aspartic acid at 419 of Eno2 is a key amino acid residue that interacts with ePgk1 to trigger the improvement of NOMN. We speculate that ePgk1 might recognize the D419 of Eno2 to trigger ePgk1-Eno2 interaction, whereas ePgk1 fails to return the same result when D419 is substituted with S419, as presented in Eno1, which we support with the reasoning given above. 

Overall, the results shown in this study afford a structural basis for understanding the multifunctional properties of Eno1 and Eno2 and their interacting proteins. In addition, these results show how even small sequence differences in proteins can lead to those proteins displaying completely different biological functions. 

## 4. Materials and Methods 

### 4.1. Plasmids Containing Various Mutated Mouse Eno2 

As illustrated in Figure 1, the recombinant mouse Eno2 fused reporter Flag (Eno2-Flag) containing DNA-encoding mutated amino acid was generated by PCR-directed mutagenesis. First, using two-step PCR, we engineered *eno2* cDNA containing the modified nucleotide(s) at the wobble (wb) position of the codon without changing the encoded amino acid residues of Eno2 (Eno2-wb). This modification avoided complete complementarity of base pairs with introduced siRNA or MO. During the first PCR, we used the flanking and mutagenic primers listed in Table 1A to amplify DNA fragments from template pCS2-mEno2-Myc [22]. Each cycle consisted of 98 °C for 30 sec for denaturation, 66 °C for 30 sec for annealing, and 72 °C for 30 sec for extension, for a total of 30 cycles. The DNA fragments were subsequently purified with a PCR extraction kit (Arrowtec) and mixed for a second round of PCR under the same conditions. The PCR-amplified fragment was inserted into pCS2 to generate pCS2-Eno2-wb-Flag (5.4 kb).

Secondly, we generated plasmids containing various point-mutated mouse Eno2, including L410I, M411L, D419S, and E420K, using flanking and mutagenic primers listed in Table 1A to amplify DNA fragments from pCS2-Eno2-wb-Flag template. Each cycle consisted of 98 °C for 30 sec for denaturation, 60 °C for 30 sec for annealing, and 72 °C for 30 sec for extension for a total of 30 cycles. The PCR-amplified fragments were inserted individually into pCS2 to generate pCS2-Eno2-wb-L410I-Flag (5.4 kb), -M411L-Flag (5.4 kb), -D419S-Flag (5.4 kb) and -E420K-Flag (5.4 kb). Regarding of plasmids of pCS2-Eno2-wb-R422K (5.4 kb), -H426R (5.4 kb) and -N427 (5.4 kb), we directly cloned by PCR using a forward flanking primer and a reverse mutagenic primer listed in Table 1B to amplify pCS2-Eno2-wb-Flag template.

### 4.2. Cell Culture

Human kidney cell line HEK293T and mouse neuroblastoma–spinal cord hybrid cell line NSC34 were used. Cell culture procedures were conducted according to the method described by Fu et al. [22], except that cells were (1) cultured in DMEM high glucose plus 10% FBS containing 1% penicillin/streptomycin and incubated at 37 °C under 5% CO_2_; and (2) subcultured when they reached 70% to 80% confluence.

### 4.3. Cell Transfection

Transfection procedures followed the standard protocol using Lipofectamine^®^ 3000 transfection reagent (Thermo Fisher Scientific). Briefly, NSC34 cells were subcultured in wells of a six-well dish, starting with a density of about 3 × 10^4^ cells per well. On the following day, we transfected 40 nM *eno2*-siRNA (Ambion, #65513, Waltham, MA, USA) for 24 h, followed by transfection with 2500 ug plasmid DNA. After 4~6 h incubation, the mixture in the wells containing DMEM medium with 2% FBS and 1 X penicillin/streptomycin to induce cell growth and differentiation was later supplemented with 2 mL serum deprived medium (SDM).

### 4.4. Purification of Recombinant Mutated Pgk1 Fused with Flag

Our experimental procedures were generally similar to those described by Fu et al. [22]. The plasmid used for transfection was pCS2-hPgk1-Flag [29]. HEK293T cells served as the host for recombinant Pgk1 protein production. They were cultured in 10 cm dishes and transformed with different plasmid constructs using JetPRIME (PolyPlus, Graffenstaden, France). The cultured cells were lysed by a whole cell extract solution containing [20 mM HEPES, 0.2 M NaCl, 1% Triton X-100, 20% glycerol, 1 mM EDTA, protease inhibitor cocktail (Roche, Basel, Switzerland), and phosphatase inhibitor (Roche)].

Neurite length developed from *eno2*-siRNA-treated NSC34 cells transfected by DNA coding for mutated Eno2 proteins

NSC34 cells were seeded in 10% FBS DMEM medium into wells of a six-well plate and incubated for 24 h. Then, cells were transfected with *eno2*-siRNA and incubated in 10% FBS DMEM medium for 24 h. After that, cells were transfected with plasmid DNA and incubated in 2% FBS for another 24 h. These plasmids included pCS2^+^, pCS2^+^-Eno2-wb-Flag, pCS2^+^-Eno2-wb-L410I-Flag, -M411L-Flag, -D419S-Flag, -E420K-Flag, -R422K-Flag, -H426R-Flag, and -N237S-Flag. After 4~6 h incubation, the mixture in the wells containing DMEM medium with 2% FBS and 1 X penicillin/streptomycin to induce cell growth and differentiation was later supplemented with 2 mL SDM. After 24 h incubation, medium was refreshed with 2%FBS until the fifth day after treatment. 

Axonal neurites derived from differentiated NSC34 cells were observed, and their neurite length was quantified as follows. Samples in a six-well plate were observed under an inverted microscope (Olympus IX71, Olympus, Tokyo, Japan), utilizing a 20× objective lens for photography. For each experimental group, 10 images were taken, and each image contained at least one cell with a neurite extending beyond 30 μm. Cells that contained at least one neurite longer than 30 μm were defined as bearing cells (in a total of 30 to 40 cells). More than 10 images were captured. The total neurite length of each bearing cell was measured using the MShot Image analysis system v1.0. The average length of neurite outgrowth per bearing cell was calculated on the fifth day of incubation. The final data of each group was averaged from three independent experiments.

b.Neurite length developed from *eno2*-siRNA-transfected NSC34 cells transfected by plasmid encoding Eno2-mutants in the presence of ePgk1

The procedures were the same as those above, except that the medium was refreshed by SDM supplemented with mouse recombinant Pgk1-Flag at a concentration of 99 ng/mL during incubation. The medium was refreshed every 24 h during NOMN derived from NSC34 cells. The average length of neurite per bearing cell was calculated on the fifth day of incubation. 

### 4.5. Zebrafish Husbandry and Use of the Transgene Line

The breeding and embryonic development stages of zebrafish were determined following the methods described by Westerfield [37] and Kimmel et al. [38], respectively. The zebrafish transgenic line *Tg(mnx1:GFP)*, in which motor neurons were tagged with green fluorescent protein, was used in the experiment [39].

### 4.6. mRNA Synthesis

To generate template DNA for mRNA synthesis, we constructed plasmids containing cDNAs of mouse *eno2-wb, eno2-wb-[D419S]*, and *eno2-wb-[E420K]*. PCR was performed using primers SP6 and T3 and the Arrow Taq DNA polymerase kit (ArrowTec, ADP0150500, Berlin, Germany) for 30 cycles. Each cycle consisted of denaturation at 95 °C for 30 sec, annealing at 55 °C for 30 sec, and extension at 72 °C for 1 min. The synthesis of 5’-capped mRNAs was conducted using the mMESSAGE mMACHINE SP6 kit (Thermo Fisher Scientific, AM1340). DNA and RNA were quantified by measuring OD260 absorbance using a spectrophotometer (DeNovix, DS-11+, Wilmington, DE, USA).

### 4.7. Rescue Experiment in Eno2-Knockdown Zebrafish Embryos

Knockdown of endogenous Eno2 in zebrafish embryos through microinjection of *eno2*-specific antisense morpholino oligonucleotides (MO) followed the protocol described by Fu et al. [22], with some modifications. First, *eno2* mRNA and its derivative mRNAs were modified at wobble-nucleotide sequences in order to avoid complete complementarity of base pairs with injected MO. Second, a mixture of 14 ng MO combined with 200 pg mRNAs was coinjected into each embryo. Third, each injection delivered 4.6 nl of the mixture. Embryos were harvested and fixed at 30 hpf.

### 4.8. Axonal Defective Phenotypes of Growing Motor Neurons Occurred in eno2-Knockdown Zebrafish Embryos Rescued by mRNA Injection 

Embryos were collected and fixed at 30 hpf and then analyzed using an inverted fluorescence microscope (Olympus IX71, objective: 20×, Olympus, Tokyo, Japan) to observe the axons from the 6th to 17th somites. A mild defect was defined as an embryo that exhibited at least one axon at one side of the trunk that was shorter than adjacent axons. A severe defect was defined as an embryo that exhibited at least one axon whose growth did not exceed the horizontal septum. The defective rate was based on the number of defective embryos out of the total number of observed embryos. The experiment was performed in triplicate, and the average values were used for Student’s *t*-test analysis.

A rescue experiment was performed by microinjection of zebrafish *eno2-wb* mRNA or its mutated *eno2-wb* mRNA into *eno2*-MO-knockdown embryos. After microinjection, the occurrence of defective phenotypes was counted and a percentage calculated. To obtain figures from whole larvae (Figure 5A,A’), a Leica M205 FCA (objective: 1×, Leica, Wetzlar, Germany) fluorescence anatomical microscope was used, while a Leica TCS SP8 (objective: 10×, Leica, Wetzlar, Germany) confocal microscope was used for obtaining detailed images (Figure 5C–E).

### 4.9. Formation of Branching Neurites from Axons in the eno2-Knockdown Zebrafish Embryos Rescued by Both mRNA Injection and Pgk1 Immersion

We followed the protocol described by Fu et al. [22], with the modification that mouse *eno2-wb, eno2-wb-[D419S],* and *eno2-wb-[E420K]* mRNAs were microinjected into the MO-treated zebrafish embryos. At 6 h post-injection, the embryos were immersed in Pgk1-Flag recombinant protein for 24 h. The neurite branching of motor neurons from 6th to 17th somites of 30-hpf embryos was then observed using an inverted fluorescence microscope (Olympus IX71, objective: 40×, Olympus, Tokyo, Japan). Neurites without branches were classified as having a normal axonal phenotype. The branched axonal phenotype was defined as neurite branching that only occurred at the central part of each axon, excluding the upper and lower fifths of each axon. The neurite branching rate was based on the number of embryos with branched neurites out of the total examined embryos. To obtain detailed images (Figure 6A,B), we used a Leica TCS SP8 (objective: 20×, Leica, Wetzlar, Germany) confocal microscope. 

### 4.10. Total Proteins Extracted from Cells and Zebrafish Embryos, Followed by Western Blot Analysis

Total proteins extracted from NSC34 cells were lysed by whole-cell extract buffer [40] containing cOmplete™ Protease Inhibitor Cocktail, EDTA-free Protease Inhibitor Cocktail (Sigma-Aldrich, St. Louis, MO, USA), and PhosSTOP (Sigma-Aldrich). After separating the proteins via 12% SDS-PAGE electrophoresis, we performed Western blot analysis using specific antibodies as follows: phosphorylated Cofilin-S3 (Cell Signaling Technology, Cat# 3313, RRID:AB_2080597; 1:1000, Danvers, MA, USA), Cofilin (Proteintech, 66057-1-Ig; 1:5000, Rosemont, IL, USA), goat anti-mouse-HRP (Proteintech, RGAM0001; 1:20,000), α-tubulin (RRID: AB_477579; 1:1000), and goat anti-rabbit-HRP (Proteintech, RGAR001; 1:20,000), which served as an internal control. 

Total protein extraction from 30-hpf zebrafish embryos was performed according to the method described by Lin et al. [40]. After extraction, protein electrophoresis was performed using 15% SDS-PAGE. The Western blot procedure followed Fu et al. [22], in which the primary antibody used was rabbit anti-Eno2 (RRID: AB_2898921; 1:1000), and the secondary antibody was goat anti-rabbit-HRP (RRID: AB_631746; 1:10,000).

Following Western blot analysis, positive bands were obtained by the iBright CL 750 Imaging System (software version: 1.8.1; Thermo Fisher Scientific, Waltham, MA, USA) and quantified using ImageJ (v1.54f).

### 4.11. Molecular Docking

The protein crystal structures used for docking experiments were obtained from the Protein Data Bank (PDB). The PDB IDs used for Pgk1 and Eno2 were 2ZGV and 5TD9, respectively.

Molecular docking predictions between Pgk1 and Eno2 were performed using ClusPro software (v2.0) [41]. PDB files of the proteins were uploaded to the ClusPro website, and the docking conditions were set by specifying the possible binding regions of the two proteins, based on our previous experimental data. The top 30 predicted structures were selected for further analysis.

Following protein-protein docking, the results were visualized using the UCSF ChimeraX software (v1.5) [42] for data interpretation and analysis. The analysis of protein surface polarity was conducted using the molecular lipophilicity potential (mlp) command in ChimeraX. In contrast, surface electrostatic potential was studied using the coulombic electrostatic potential (coulombic) command. 

### 4.12. Statistical Analysis

For statistical analysis, we used one-way ANOVA, followed by either Tukey’s multiple comparison test or Student’s *t*-test for statistical analysis. The analyses were performed with GraphPad Prism v9, with significance determined at the *p*-value indicated in the figure legend. The calculation of the F-value was obtained during statistical analysis using GraphPad Prism v9.

## Figures and Tables

**Figure 1 ijms-25-10753-f001:**
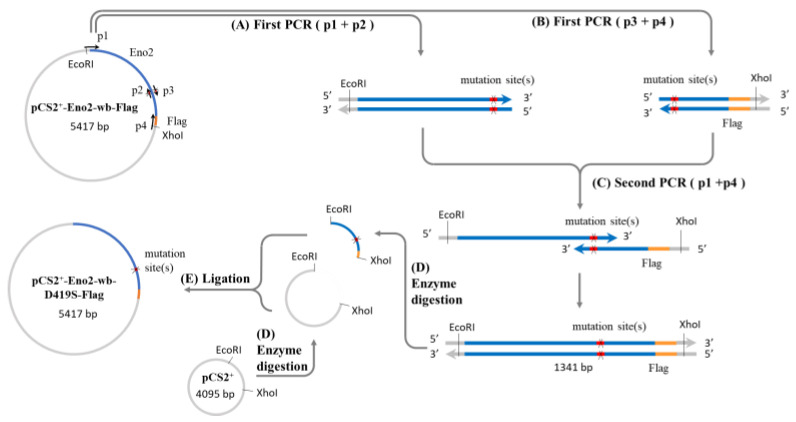
Construction of Eno2 mutant protein expression plasmid. A two-step PCR method was used to generate the expression plasmid, with the mutation of aspartic acid (D) to serine (S) at position 419 (Eno2-[D419S]) as an example. Four primers (p1 to p4) were used for the PCR, where p1 and p4 represent the forward and reverse flanking primers of the *eno2* fragment, and p2 and p3 are complementary mutagenic primers carrying the mutation sequence. The plasmid pCS2-Eno2-wb- Flag containing the mouse *eno2* gene fused with a Flag reporter gene and with wobble-modified nucleotides (wb) was used as the template in the first PCR step. Primers p1 and p2 were used to produce the upstream fragment of the gene containing the mutation site (X), while p3 and p4 were used to produce the downstream fragment. In the second PCR step, the products from the first step were mixed, and a small amount of p1 and p4 primers were added to generate the complete mutant gene fragment. After PCR completion, the mutant gene fragment and the pCS2 plasmid were digested with EcoRI and XhoI restriction enzymes to produce sticky-ended inserts and vectors. Finally, the insert and vector were ligated to create the pCS2-derived expression plasmid for the Eno2 mutant protein fused with the Flag reporter gene. Blue color: *eno2* cDNA; orange color: reporter Flag; red color: mutation site; grey color: plasmid pCS2^+^ backbone.

**Figure 2 ijms-25-10753-f002:**
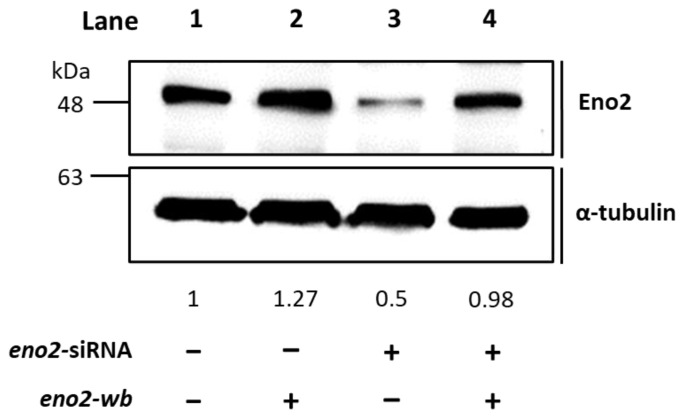
Western blot analysis demonstrated that the recombinant Eno2-wb protein was expressed in the Eno2-knockdown cells transfected with siRNA. *NSC34* cells were transfected with *eno2* siRNA to knock down the endogenous Eno2, followed by transfection DNA encoding Eno2 but consisting of nucleotides modified at the wobble position (Eno2-wb) to avoid the block by introducing siRNA. The expression level of Eno2 protein in the cells was detected by the Eno2 antibody. Lane 1 was the control group, and its Eno2 expressional level was set at 1, while lanes 2–4 represented Eno2 expressed within NSC cells treated without (−) or with (+) *eno2* siRNA or *eno2-wb*. The expressional level of Eno2 was quantified relative to that of α-Tubulin that served as an internal control. The number listed below each lane indicates the fold change of Eno2 intensity of each treatment compared to the control group, set as 1.

**Figure 3 ijms-25-10753-f003:**
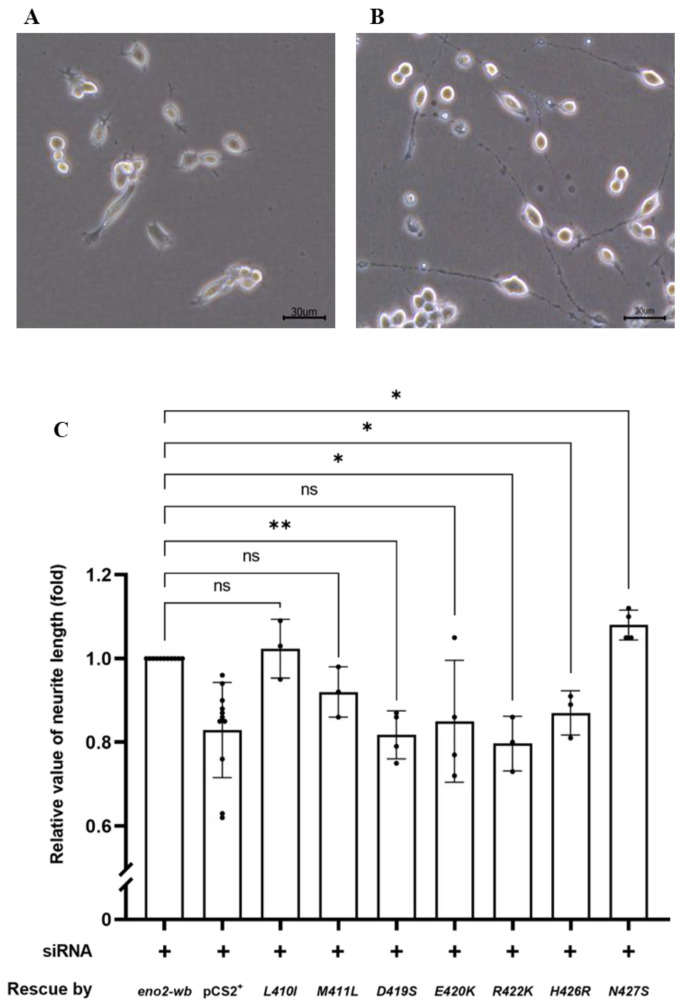
The effect of various point mutations in Eno2 on promoting the neurite growth of motor neurons. (**A**,**B**) Morphological differentiation of cultured NSC34 neural cells: (**A**) one-day incubation; (**B**) five-day incubation. Axonal neurites derived from cultured NSC34 cells were observed, and the neurite length was measured. (**C**) Statistical analysis of the average length of neurites. NSC34 cells transfected with siRNA (pCS2^+^) to inhibit endogenous mouse *eno2* served as a negative control. After transfection with *eno2* siRNA, *eno2* DNA with modified wobble-nucleotides (*eno2-wb*) was transfected in NSC34. The resultant neurite length was normalized to a value of 1.0, which served as the positive control. The average neurite length of motor neurons was measured after transfection with siRNA, followed by transfection with plasmid harboring different point-mutated *eno2-wb* DNA. The fold change in neurite length was calculated relative to the positive control set as 1.0. Then, each experimental group was independently compared with the control group using *t*-test for statistical analysis (* *p* < 0.05, ** *p* < 0.01; ns indicates no significant difference). The *t* values and degrees of freedom for groups of *-[L410I]*, *-[M411L]*, *-[D419S]*, *-[E420K], -[R422K], -[H426R]* and *-[N427S]* were 0.5754 and 2, 2.309 and 2, 6.362 and 3, 2.064 and 3, 5.413 and 2, 4.255 and 2, and 4.496 and 3, respectively.

**Figure 4 ijms-25-10753-f004:**
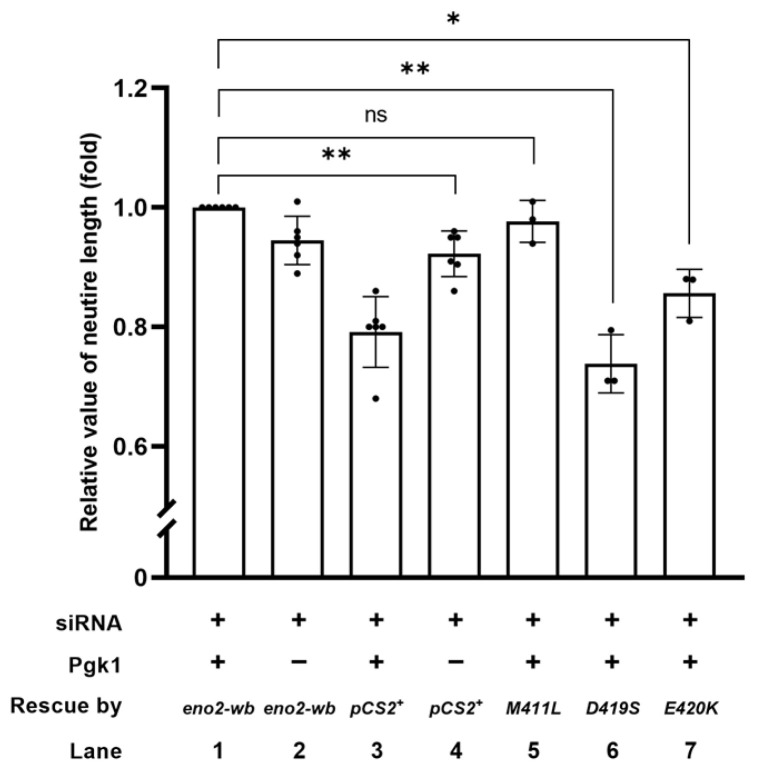
The synergistic effect of promoting motor neurons by the addition of Pgk1 and transfection of various mutations of Eno2. Statistical analysis of the average length of neurites. NSC34 neural cells transfected with siRNA (pCS2+) to inhibit endogenous mouse Eno2 without the addition of extracellular Pgk1 (ePgk1) served as a negative control. As the positive control, NSC34 cells were transfected with *eno2* siRNA, followed by transfection of wobble-modified *eno2* DNA (*eno2-wb*) combined with Pgk1 immersion. The resultant average neurite length was normalized as 1.0. Similarly, plasmid harboring a point-mutated *eno2* DNA, as indicated, was transfected in siRNA-treated cells, followed by Pgk1 immersion. The average neurite length of motor neurons was measured, and fold change compared to the positive control was calculated. Each experimental group was independently compared with the control group using Student’s *t*-test for statistical significance (* *p* < 0.05; ** *p* < 0.01; ns indicates no significant difference). The t values and degrees of freedom for groups of pCS2^+^, *-[M411L]*, *-[D419S]* and *–[E420K]* were 4.972 and 5, 1.151 and 2, 9.281 and 2, and 6.189 and 2, respectively.

**Figure 5 ijms-25-10753-f005:**
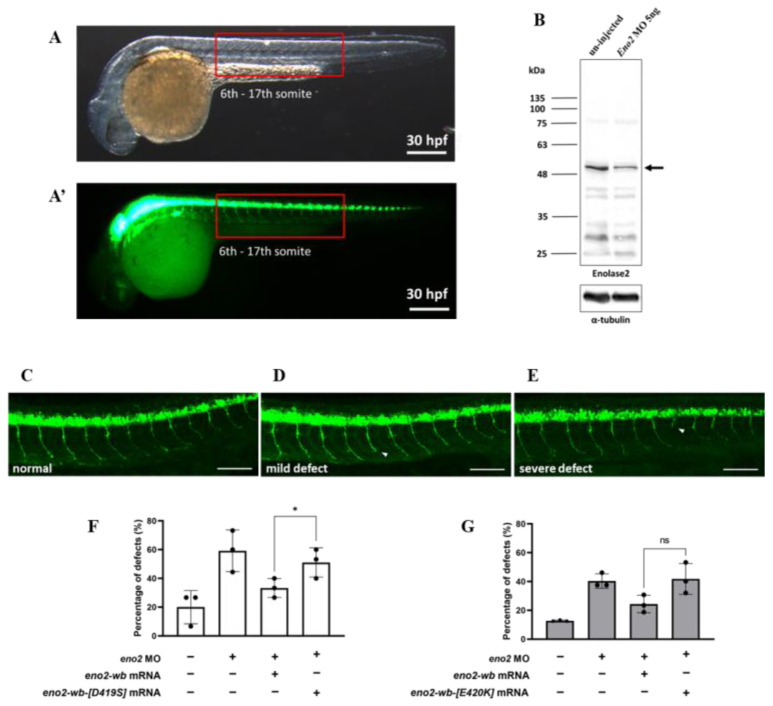
The rescue effect of mutant *eno2* mRNA on Eno2 knockdown in zebrafish embryos. Microinjection was performed at the one-cell stage in the transgenic line *Tg(mnx1:GFP)*. Embryos were collected at 30 hpf and observed for motor neurons in the 6th to 17th somites, as shown in Figure (**A**,**A’**) with (**A**) under visible light and (**A’**) under fluorescence. (**B**) Western blot analysis comparing Eno2 protein levels (indicated by arrows) in embryos with and without MO injection, using rabbit anti−Eno2 antibodies for detection. (**C**–**E**) Three phenotypes: normal, mild defect, and severe defect with arrows indicating defect locations. Scale bar: 100 μm. (**F**,**G**) Frequency of defects. Control groups were designated as (1) the control group without treatment; (2) the eno2 MO group in which injection with eno2-specific antisense morpholino oligonucleotide (MO) served as a negative control; and (3) the MO plus *eno2* mRNA group in which co-injection of eno2 MO and *eno2*-wb mRNA served as a positive control. Experimental groups were (**F**) the MO plus *eno2-wb-[D419S]* mRNA group: co-injection of eno2 MO combined with *eno2-wb-[D419S]* mRNA; and (**G**) the MO plus *eno2[E420K]* mRNA group: co-injection of eno2 MO combined with *eno2-wb-[E420K]* mRNA. Both figures represent the averages of three independent experiments. White indicates the percentage of severe defects, gray indicates the percentage of mild defects, and black indicates the overall defect percentage. Statistical significance was analyzed using one-way ANOVA (* indicates *p* < 0.05; ns indicates no significant difference). F statistics of (**F**) were 1.401 as numerator and 2.802 as denominator, while (**G**) were 1.917 as numerator and 3.833 as denominator.

**Figure 6 ijms-25-10753-f006:**
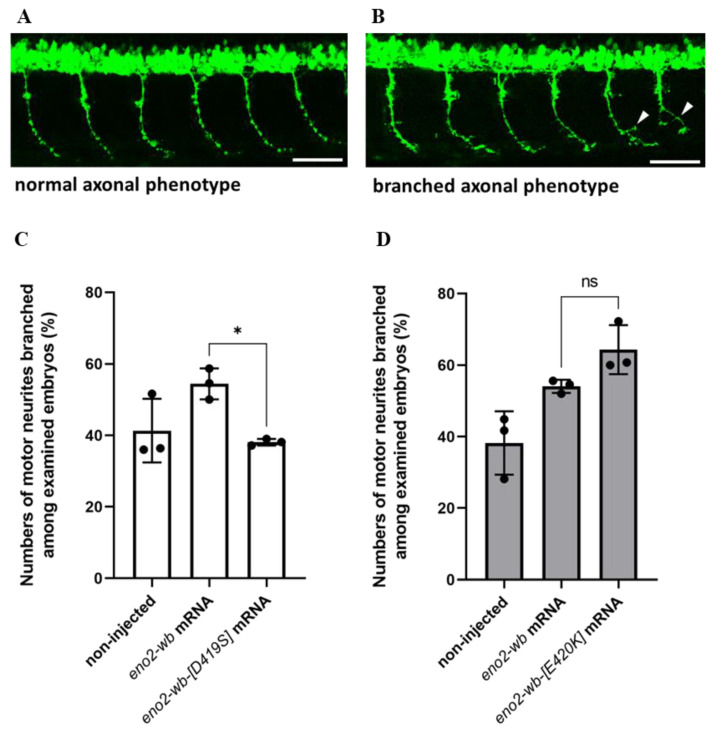
The effect of overexpressing mutant Eno2 on the occurrence rate of branched axons of motor neurons in zebrafish embryos immersed with recombinant Pgk1. (**A**,**B**) Two phenotypes of the caudal primary (CaP) axons of motor neurons in the 30 hpf transgenic line *Tg(mnx1:GFP)* embryos. (**A**) Examples of normal phenotype and (**B**) branched axon phenotype (branching sites indicated by white arrowheads). Scale bar: 50 μm. Experimental manipulations were performed at the one-cell stage. The first group of zebrafish embryos served as the non-injected control group, the second group was injected with mouse *eno2-wb* mRNA, and the third group was injected with mutated *eno2-wb* mRNA. At 30 hpf, the percentage of embryos exhibiting the branched axonal phenotype among CaP neurons was calculated for each group. (**C**) The results of embryos injected with *eno2-wb-[D419]* mRNA and (**D**) the results of embryos injected with *eno2-wb-[E420K]* mRNA. Each panel represents the average of three independent experiments with statistical analysis performed using one-way ANOVA (*, *p* < 0.05; ns indicates no significant difference). F statistics of (**C**) were 1.239 as numerator and 2.478 as denominator, while (**D**) were 1 as numerator and 2.001 as denominator.

**Figure 7 ijms-25-10753-f007:**
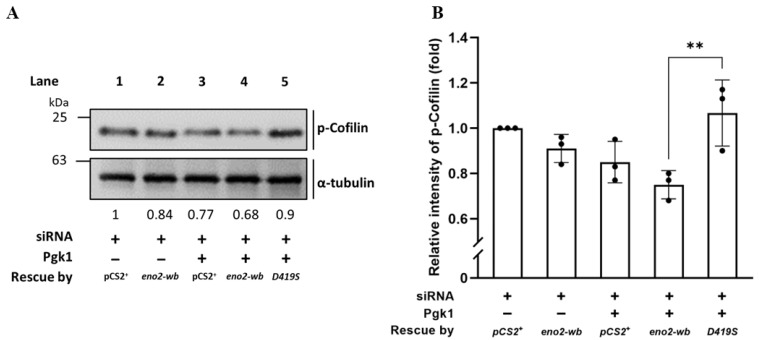
Using Western blot to analyze the level of phosphorylated Cofilin (p-Cofilin) expressed in NSC34 cells. (**A**) Western blot analysis. NSC34 cells were transfected with *eno2*-siRNA to knock down endogenous Eno2, then rescued with different DNA material, as indicated, in the absence (−) or presence (+) of Pgk1 protein. pCS^+^: plasmid with an siRNA insertion; *eno2-wb*: wild type Eno2 but consisting of nucleotides modified at wobble position (*eno2-wb*) to avoid being blocked by introducing siRNA; and *D419S*: mutant Eno2-wb consisting of an aspartic acid mutated into serine at the 419th position of Eno2. The expressional level of p-Cofilin at Ser3 was quantified relative to that of α-Tubulin, which served as an internal control. The number listed below each lane indicates the fold change of the p-Cofilin intensity of each treatment compared to the control group, set as 1. (**B**) Quantitative and statistical analyses. Data were averaged from three independent experiments and presented as mean ± SD (*n* = 3). One-way ANOVA, followed by Tukey’s multiple comparison test, was used to perform statistical analysis (** *p* < 0.005). F statistics were 4 as numerator and 10 as denominator.

**Figure 8 ijms-25-10753-f008:**
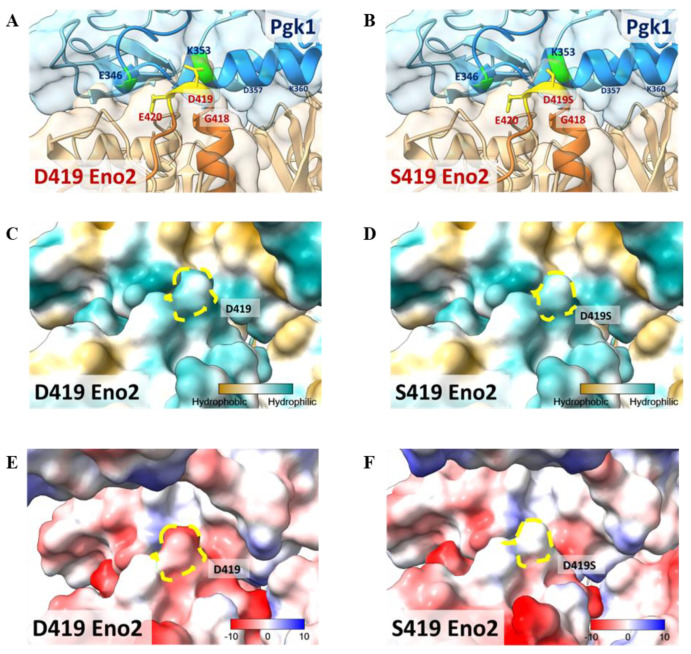
Molecular docking model to illustrate the key amino acid involved in Eno2-ePgk1 interaction. (**A**,**B**) Simulated model to illustrate how Eno2 (PDB ID: 5TD9) interacted with ePgk1 (PDB ID: 2ZGV) is presented in blue and orange, respectively. Drawing of partial enlargement of critical amino acid residues involved in Eno2-ePgk1 interaction was presented, including (**A**) wild-type Eno2 D419 and (**B**) mutant Eno2 D419S. The segment of the 345th to 360th amino acids of Pgk1, the segment of the 404th to 431st amino acid of Eno2, and the mutant Eno2 D419th were shown in Dodger blue, chocolate, and orange, respectively. (**C**,**D**) Drawing of partial enlargement of the surface polarity distribution of (**C**) Eno2 D419 and (**D**) Eno2 D419S. Dotted circles highlighted the D419th residue on Eno2. Lipophilicity was presented as low (hydrophilicity, in cyan) to high (hydrophobicity, in brown). Dotted zone indicates the 419th site. (**E**,**F**) Drawing of partial enlargement of the surface charge distribution of (**E**) Eno2 D419 and (**F**) Eno2 D419S. Chargeability from negative charge (in red) to positive charge (in blue). Dotted circles marked the 419th residue of Eno2.

**Table 1 ijms-25-10753-t001:** The primers used to generate mutated Eno2.

Primer Name	Sequence (5’ → 3’)
(A) Primers used to generate Eno2-wb and Eno2 mutants: L410I, M411L, D419S, E420K	
flanking primers	
mEno2-EcoRI F	TATATGAATTCATGTCTATAGAGAAGATTTGGG
mEno2-Flag -XhoI R	TATATCTCGAGTCACTTATCGTCGTCATCCTTGTAATCCAGCACACTGGGATTTC
mutagenic primers	
mEno2-wobble F	GACCCATCACGCTATATAACTGGGGACCAGCTG
mEno2-wobble R	TTATATAGCGTGATGGGTCAGCGGGAGACTTG
mEno2-L410I F	AAGTACAACCAGATCATGAGAATTGAG
mEno2-L410I R	CTCAATTCTCATGATCTGGTTGTACTT
mEno2-M411L F	TACAACCAGCTCCTCAGAATTGAGGAA
mEno2-M411L R	TTCCTCAATTCTGAGGAGCTGGTTGTA
mEno2-D419S F	GAGGAAGAGCTGGGGAGCGAAGCTC
mEno2-D419S R	GAGCTTCGCTCCCCAGCTCTTCCTC
mEno2-E420K F	GCTGGGGGACAAAGCTCGCTTCG
mEno2-E420K R	CGAAGCGAGCTTTGTCCCCCAGC
(B) Primers used to generate Eno2 mutants: R422K, H426R, N427K	
flanking primers	
mEno2-EcoRI F	TATATGAATTCATGTCTATAGAGAAGATTTGGG
mutagenic primers	
mEno2-R422K R	AACTCGAGTCACAGCACACTGGGATTTCGGAAATTATGTCCCGCGAACTTAGCTTCGTCCCC
mEno2-H426R R	AACTCGAGTCACAGCACACTGGGATTTCGGAAATTCCTTCCCGCGAACTT
mEno2-N427S R	AACTCGAGTCACAGCACACTGGGATTTCGGAAGGAATGTCCCGCGAA

The mutated site is indicated by a box. The cutting site of restriction enzyme is underlined.

## Data Availability

The original contributions presented in the study are included in the article, further inquiries can be directed to the corresponding author.

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
