# Peer review of "The 419th Aspartic Acid of Neural Membrane Protein Enolase 2 Is a Key Residue Involved in the Axonal Growth of Motor Neurons Mediated by Interaction between Enolase 2 Receptor and Extracellular Pgk1 Ligand"

_ijms, 2024, doi:10.3390/ijms251910753_

Round 1

Reviewer 1 Report

Comments and Suggestions for Authors

In the current manuscript by Lee et al., the authors investigated which amino acid residue of neuronal Enolase (Eno2) is required for the interaction with extracellular Pgk1 (ePgk1) and the neurite outgrowth of motor neuron (NOMN). They identified the D419 of Eno2 is the key residue for MOMN. The finding is interesting and may be worth publishing, the current version is not acceptable because of the lack of statistical rigor.

1. The introduction is redundant. The “The Enolase family”, “Moreover, Eno2,”, and “Amyotrophic lateral sclerosis (ALS)” sections may be combined into a single section. The contents of this manuscript only relate to Eno2 and not to ALS, the introduction may be focused on Eno2 and need not mention to ALS.

2. Fig.2, “kDa” is missing.

3. Figs. 3C and 4, please do not use the t-test, and please use appropriate multiple comparison such as the Dunnett, Bonferroni, Tukey test after ANOVA.

4. Figs. 5F, 5G, 6C, and 6D, again please do not use the t-test for multiple comparison.

5. Fig. 7 needs to be statistically analyzed after densitometric analysis. It is not sure, for example, whether the reduction to 0.84 was actually significant.

6. In the discussion, the authors compared the structural difference between Eno1 and Eno2, can the authors provide molecular docking of Eno1 and ePgk1?

Author Response

In the current manuscript by Lee et al., the authors investigated which amino acid residue of neuronal Enolase (Eno2) is required for the interaction with extracellular Pgk1 (ePgk1) and the neurite outgrowth of motor neuron (NOMN). They identified the D419 of Eno2 is the key residue for MOMN. The finding is interesting and may be worth publishing, the current version is not acceptable because of the lack of statistical rigor.

(1) The introduction is redundant. The “The Enolase family”, “Moreover, Eno2,”, and “Amyotrophic lateral sclerosis (ALS)” sections may be combined into a single section. The contents of this manuscript only relate to Eno2 and not to ALS, the introduction may be focused on Eno2 and need not mention to ALS.
Author’s response to (1):

In accordance with your suggestion, we reorganized the “The Enolase family,” “Moreover, Eno2,” and “Amyotrophic lateral sclerosis (ALS)” into a single section and focused on the description of Eno2. (please see lines 48-87)

(2) Fig.2, “kDa” is missing.
Author’s response to (2):

Thank you for your reminder. We added kDa in Figs. 2 and 7.

(3) Figs. 3C and 4, please do not use the t-test, and please use appropriate multiple comparison such as the Dunnett, Bonferroni, Tukey test after ANOVA.
Author’s response to (3):

We’re apology for not being described clearly in Figs. 3C and 4. What we performed statistical analyses of the results shown on these two figures were designed under the condition that each experimental group was independently compared with the control group. Therefore, we believe that the Student’s t-test should be an appropriate method applied to perform statistical analysis. We added this detailed statement in the legends of these two figures.

(4) Figs. 5F, 5G, 6C, and 6D, again please do not use the t-test for multiple comparison.
Author’s response to (4):

Thank you for pointing out this mistake. We re-analyzed the data and used the Tukey test in Figs. 5F, 5G, 6C, and 6D. We revised this description in the Materials and methods (lines 684-687) and in the legends of these figures (lines 288-290, 329-331).

(5) Fig. 7 needs to be statistically analyzed after densitometric analysis. It is not sure, for example, whether the reduction to 0.84 was actually significant.
Author’s response to (5):
In accordance with your comment, we conducted statistical analysis from three independent experiments and showed in the newly added Fig.7B. The result demonstrated that there was a significant difference between eno2-wb- and eno2-wb-D419 groups.

(6) In the discussion, the authors compared the structural difference between Eno1 and Eno2, can the authors provide molecular docking of Eno1 and ePgk1?
Author’s response to (6):

We apology for not being able to provide this model since we do have a different point of view. Fu et al. (2023) reported that only Eno2 interacts with Pgk1, not Eno1, to promote motor neuron growth. Therefore, in this study, we attempted to employ molecular docking analysis to obtain a deeper insight into the specific binding sites and structural characteristics of the interactions between Pgk1 and Eno2. The premise of molecular docking is typically based on the assumption that there is potential binding capability or interaction between the molecules. However, performing molecular docking may be quite limited meaning for two proteins without being known their interaction (e.g., Pgk1 and Eno1), although the software may still have chance to predict some "theoretical" binding sites. Yet, the biological relevance of these sites may be minimal or nonexistent.

Reviewer 2 Report

Comments and Suggestions for Authors

In this paper, the authors use several different outcome measures in vitro and in vivo to show the reason for specificity of the interaction between Eno2 and ePgk1, which underlies outgrowth of motor neurons.

Thus, the authors shed light on why Eno1 does not bind ePgk1, despite the fact that Eno1 and Eno2 are highly homologous.

The data are very complimentary and present a good story.

Nevertheless, I have some questions I would like the authors to consider.

Citations are numbered in the bibliography and are name year in text. Please justify. Lin et al., 2019 a and b - please add a or b to the bibliography.

Figures 2 and 7.

For Western blotting - please provide full quantification with variability within each condition.

For all graphs
1) Unfortunately, column graphs are no longer acceptable. Please provide scatter plots with superimposed errors (please define the errors), or box+whisker plots with median and means shown (please ensure to define the whiskers). Or, perhaps the easiest would be to superimpose the mean of individual experiments on top of the columns.
2) Please state the number of biologic replicates (number of independent experiments) and the number of technical replicates within each experiment.
3) Please add the error bars to the control to show the variability within this group.
4) For all ANOVAs, please provide F statistics together with numerator and denominator. For all ttests, please provide t values and degrees of freedom.

Fig 3 C
What conditions were significantly different from control (asterisks are not apparent)?
Why is there no variability in the H426R group?

Figure 6
Please justify how these images were of sufficient resolution to view the branching, for e.g., A appears much more grainy (less resolved) than B.

Methods
Section 4.8 - what objective was used?
These data should be analysed using 1-way ANOVAs (not a ttest).

Section 4.9
Please detail the imaging conditions (microscope, objective, resolution, whether it was confocal or epifluorescence, etc)

Author Response

In this paper, the authors use several different outcome measures in vitro and in vivo to show the reason for specificity of the interaction between Eno2 and ePgk1, which underlies outgrowth of motor neurons.

Thus, the authors shed light on why Eno1 does not bind ePgk1, despite the fact that Eno1 and Eno2 are highly homologous.

The data are very complimentary and present a good story.

 Nevertheless, I have some questions I would like the authors to consider.

(1) Citations are numbered in the bibliography and are name year in text. Please justify. Lin et al., 2019 a and b - please add a or b to the bibliography.
 Author’s response to (1):

Thank you for pointing out this mistake. We revised it as following: The plasmid used for transfection was pCS2-hPgk1-Flag (Lin et al., 2019a) (please see line 563). We also added Lin et al. (2019a) to the bibliography (please see lines 767-769).

(2) Figures 2 and 7.
For Western blotting - please provide full quantification with variability within each condition.
 Author’s response to (2):

In response to your comment, we provided full quantification (please see lines 369-372, and Fig. 7B).

(3) For all graphs
(3A) Unfortunately, column graphs are no longer acceptable. Please provide scatter plots with superimposed errors (please define the errors), or box+whisker plots with median and means shown (please ensure to define the whiskers). Or, perhaps the easiest would be to superimpose the mean of individual experiments on top of the columns.
Author’s response to (3A):

Thank you for your suggestion. We added newly revised figures in which scatter plots with superimposed the standard deviations for each group on top of the columns.

(3B) Please state the number of biologic replicates (number of independent experiments) and the number of technical replicates within each experiment.
Author’s response to (3B):

In response to your suggestion, we described in more detailed and added in the Materials and Methods as follows: Regarding quantification of NOMN derived from cultured neural cells, 3x104 NSC34 neural cells were cultured in each group and observed under microscope. Each trial, we randomly chose ten 20X microscopic pictures exhibiting at least one cell with a single neurite longer than 30 μm, collected them in a total of 30 to 40 cells and measured their length of neurite outgrowth. The final data of each group was averaged from three independent experiments (Please see lines 556, 583-590).

(3C) Please add the error bars to the control to show the variability within this group.
Author’s response to (3C):

After analyzing the results from each experiment, we normalized the value of control group as 1 before comparing the different groups. Therefore, the control group did not have error bars.

(3D) For all ANOVAs, please provide F statistics together with numerator and denominator. For all ttests, please provide t values and degrees of freedom.
 Author’s response to (3D):

In response to your suggestion, we added F statistics in the legends of Figs. 5,6 and 7, while we added t values and degrees of freedom in the legends of Figs. 3 and 4. Additionally, we provided all information obtained from Statistical analyses in the supplementary data as follows.

(4) Fig 3 C
(4A) What conditions were significantly different from control (asterisks are not apparent)?
Author’s response to (4A):
We conducted statistical analysis, and the results indicated a significant difference for D419S from control was two-asterisk level (please refer to Fig. 3C).

(4B) Why is there no variability in the H426R group?
 Author’s response to (4B):

Thank you for pointing out. We added the statistical analysis results from three independent experiments for the H426R group in the revised version (please refer to Fig. 3C).

(5) Figure 6
Please justify how these images were of sufficient resolution to view the branching, for e.g., A appears much more grainy (less resolved) than B.
 Author’s response to (5):

Thank you for your comment. We replaced them with high-resolution images (please refer to revised Fig. 6).

(6) Methods
(6A) Section 4.8 - what objective was used?
Author’s response to (6A):

We added the objective lens specifications in the revised text (please refer to line 624 and lines 633-636).

(6B) These data should be analysed using 1-way ANOVAs (not a ttest).
Author’s response to (6B): 

Thank you for pointing out this mistake. We re-analyzed the data using the Tukey test in Figs. 5F, 5G, 6C, and 6D, and revised them.

(6C) Section 4.9
Please detail the imaging conditions (microscope, objective, resolution, whether it was confocal or epifluorescence, etc)
Author’s response to (6C):

In response to your suggestion, we specified the imaging conditions in more detail in the revised manuscript as follows:

(1) At 6 hr post-injection, ...using an inverted fluorescence microscope (Olympus IX71, objective: 40x) (please see lines 643-644); and

 (2) To obtain detailed images (Figs. 6A, B), we employed Leica TCS SP8 (objective: 20x) confocal microscope (please see lines 648-649).

Round 2

Reviewer 1 Report

Comments and Suggestions for Authors

The manuscript has been improved. I have no further comment.

Reviewer 2 Report

Comments and Suggestions for Authors

I thank the authors for their careful and detailed responses, which address all of my comments.